# Optimizing Motor Imagery Parameters for Robotic Arm Control by Brain-Computer Interface

**DOI:** 10.3390/brainsci12070833

**Published:** 2022-06-26

**Authors:** Ünal Hayta, Danut Constantin Irimia, Christoph Guger, İbrahim Erkutlu, İbrahim Halil Güzelbey

**Affiliations:** 1Pilotage Department, Faculty of Aeronautics and Aerospace, Gaziantep University, 27310 Gaziantep, Turkey; 2Department of Energy Utilization, Faculty of Electrical Engineering, Electrical Drives and Industrial Automation (EUEDIA), “Gheorghe Asachi” Technical University of Iasi, 700050 Iași, Romania; 3g.tec Medical Engineering GmbH, 4521 Schiedlberg, Austria; guger@gtec.at; 4Brain and Nerve Surgery Clinic, Liv Hospital Gaziantep, 27080 Gaziantep, Turkey; ibrahimerkutlu@gmail.com; 5Faculty of Aeronautics and Aerospace, Hasan Kalyoncu University, 27010 Gaziantep, Turkey; guzelbeyih@hku.edu.tr

**Keywords:** Brain-Computer Interface, EEG, motor imagery, common spatial patterns (CSP), robot control

## Abstract

Brain-Computer Interface (BCI) technology has been shown to provide new communication possibilities, conveying brain information externally. BCI-based robot control has started to play an important role, especially in medically assistive robots but not only there. For example, a BCI-controlled robotic arm can provide patients diagnosed with neurodegenerative diseases such as Locked-in syndrome (LIS), Amyotrophic lateral sclerosis (ALS), and others with the ability to manipulate different objects. This study presents the optimization of the configuration parameters of a three-class Motor Imagery (MI) -based BCI for controlling a six Degrees of Freedom (DOF) robotic arm in a plane. Electroencephalography (EEG) signals are recorded from 64 positions on the scalp according to the International 10-10 System. In terms of the resulting classification of error rates, we investigated twelve time windows for the spatial filter and classifier calculation and three time windows for the variance smoothing time. The lowest error rates were achieved when using a 3 s time window for creating the spatial filters and classifier, for a variance time window of 1.5 s.

## 1. Introduction

Humans use peripheral nerves and muscles to control and communicate with the environment such as operating robots. This process includes complex steps between the brain, peripheral nervous system, and the corresponding muscles which are responsible for communication and control. In an efferent pathway, signals are generated in the brain and are sent to muscles over the peripheral nervous system. In contrast, impulses are conveyed from sensory receptors to the central nervous system in an afferent pathway [1]. Disorders such as amyotrophic lateral sclerosis (ALS), brainstem stroke, and spinal cord injury can disrupt these pathways and cause loss of voluntary muscle control as a result of which humans cannot use their arms, legs, and even faces. Therefore, a fully operational mind is locked in the body and cannot communicate with the environment [2].

Among the methods which help to restore the communication and control functions of severely disabled people, BCI is one of the most powerful options. A BCI replicates the natural communication system of the body. In place of the peripheral nerves and muscles, a BCI directly records activities in the brain related to the user’s intent and converts them to control signals for the application. Some standards exist for a BCI. Firstly, recordings must be taken directly from the brain. Real-time feedback must be given to the user, and the system must depend on voluntary control. The BCI system is composed of four main parts: acquisition of signal, feature extraction, classification, and output control commands [3].

Brain activity generates electrical and magnetic variation over different parts of the brain. These activities can be measured by sensors either invasively or noninvasively. The invasive methods have advantages such as good signal quality, high spatial resolution, and a high-frequency range. However, they also have some serious drawbacks like needing surgery, continuous connection to the device, and tissue reactions to the device. In comparison, the noninvasive techniques have several advantages such as the absence of surgical risk, high temporal resolution, inexpensiveness, and high signal stability. EEG is the most preferred non-invasive signal acquisition technique for BCIs. This method can measure any change in brain activity within milliseconds and offers lower spatial resolution than invasive methods, yet it can record up to 256 channels simultaneously.

The noninvasive BCI applications can be divided into three main approaches. The selection of the approach depends on the requirements of the application. For example, if a great number of discrete classes will be controlled, P300 evoked potentials is the best result [4,5]. Another approach is steady-state visual evoked potential (SSVEP). Robotic applications [6], and prosthetic device control [7] can be done through this type of paradigm. Both of these methods need modest training time and can be easily used by a large number of people. The last method is the motor imagery-based BCI. This approach requires more user training and has lower accuracy in comparison to P300 and SSVEP. An external stimulus is required for P300 and SSVEP. On the other hand, motor imagery (MI)-based BCI does not require any external stimulation and it can be used endogenously [8,9]. Only stimulation in the system is used for feedback to the user. The feedback can be used for operant conditioning.

Actual movement and the imagination of a movement make a change in the brain’s oscillatory activity. These oscillations are named the sensorimotor rhythms (SMRs). SMRs are composed of signals with different frequencies. In SMR-based BCI applications, alpha oscillations (7–13 Hz) also referred to as mu rhythms, and beta oscillations (13–30 Hz) are mostly used [10]. There are two kinds of modulations in the oscillatory activity. Suppression of the oscillation is called event-related desynchronization (ERD). This decrease in the amplitude of the rhythm is produced during actual movement execution or motor imagery [11]. In contrast, the raise of rhythmic activity is called event-related synchronization (ERS) [12]. By using these modulations, two-class MI-based BCIs have been developed [9,13,14].

In addition, studies including three [8] and four classes [15] are available in the literature. To measure these modulations accurately, the electrodes are placed over the sensorimotor area of the brain. It has been claimed that a single EEG trial classification accuracy of 80–95% can be achieved using two electrodes over the C3 and C4 [9]. However, two bipolar electrodes are not sufficient to describe whole brain activity. Therefore, using more EEG signals over the sensorimotor area can enhance classification accuracy. Moreover, placing electrodes over premotor and supplementary motor areas can supply information for the separation of the brain states which is related to motor imagery [16].

The next stage of the BCI is signal processing. Several techniques are used for the extraction of features from the EEG data. Common spatial patterns (CSP) is a more powerful technique for extracting information than other spatial filters such as bipolar, Laplacian, and common average reference (CAR) [16,17,18]. In recent years, researchers developed many extensions of the CSP algorithm to differentiate between two conditions better. For example, the invariant common spatial patterns (iCSP) [19] was developed to address the nonstationarities in the recording stage; the common spatio-temporal pattern (CSTP) analysis [20] that uses the time embedding method and concatenates the features extracted from each time-windowed interval; CSP method that uses the Riemannian geometry [21], and many other. For classification analysis, linear discriminant analysis (LDA), support vector machine (SVM), k-nearest neighbor (kNN), etc. [22,23] can be used, among which, LDA is preferred more due to its performance and low computational requirements [24].

The control of a robotic arm or an assistive robot with a non-invasive BCI provides a desirable alternative for disabled people. The concept of the assistive device control has been proposed in previous research and explored in offline analyses or online cases including control of a virtual reality avatar [25,26,27], wheelchairs, quadcopters [28,29], and various other rehabilitation devices successfully [30,31]. One of the most important parts of device control via BCI is classification accuracy. Regardless of the mental strategy used, the classification accuracy must go above 80% (or below 20% in terms of classification error rate). For healthy subjects, it is easier to exceed this value than for disabled people whose medical conditions may cause a decrease in the control accuracy. Due to this fact, it is very important to check different ways to increase the BCI classification accuracy.

In this study, an MI-based BCI system was developed to control a 6 degrees of freedom robotic arm in a plane. The whole system is composed of two subsystems: the first is the BCI system and the other is the robot control unit. The BCI system acquires the brain signals from the human brain which are related to mental states such as left-hand, right-hand, and foot movement imagination. These signals are converted into corresponding control signals like left, right, and forward and sent to the robot. To achieve a better classification accuracy, we investigated twelve epochs off-line to create the spatial filters, a classifier and three different time window lengths for the variance smoothing time.

## 2. Materials and Methods

### 2.1. Data Recording

Seven healthy subjects participated to our study. During the experiments, 64 EEG electrodes were placed over the primary sensorimotor area in accordance with the international 10-20 system [32]. The reference electrode for achieving the bipolar derivation for all EEG channels was placed on the right ear and the ground electrode was placed on the forehead (Fz). The electrode placements are illustrated in Figure 1. The EEG signal was recorded using g.HIamp biosignal amplifier and g.Hisys software V3.16.00 (g.tec medical engineering GmbH, 4521 Schiedlberg, Austria) [33]. All signals were amplified and recorded. The signals were sampled with a rate of 256 Hz and filtered at 0.5 Hz- 30 Hz 8th order Butterworth band-pass filter. To cancel power line noise, a Notch filter with 48–52 Hz was applied.

### 2.2. Experimental Paradigm

The subject was seated one meter away from the computer. At the beginning of the experiment, a fixation cross was shown to the subject as demonstrated in Figure 2. After two seconds, a warning tone was heard. This warning tone informed the subject that the trial had started. Between the third and fourth seconds, a cue, shown in the form of left, right, and forward arrows, was presented. The subject tried to imagine movement corresponding to these arrows. Between the fourth and eighth seconds, the real-time EEG signal was recorded. After the eighth second, there was time for relaxation until the next trial started. These relaxation times were randomly varied between 0.5 and 2.5 s.

Each cue was shown 15 times (left, right, forward) which amounts to 45 trials for each run. No feedback was provided to the user during training runs. Every subject repeated the training runs four times. At the end of the experiment, one subject produced 180 motor imagery trials. This training data of four runs were recorded using the CSP filters and the classifier from the previous session. By using new training data, CSP filters and an LDA classifier were generated for the new session. In the test run, visual feedback and robot movements were provided to the user as feedback. One session eventually contained 4 training and 2 test runs as shown in Figure 3. The MATLAB/Simulink model for the motor imagery-based BCI control of the robotic arm is shown in Figure 4.

### 2.3. Feature Extraction and Classification

The common spatial patterns (CSP) method is a frequently used algorithm for Motor Imagery discrimination tasks. It uses covariance to design common spatial patterns and is based on the simultaneous diagonalization of two covariance matrices [16]. The method maximizes the variance for one class while minimizing it for the other one [34]. By doing so, the difference between the populations of the two classes is maximized, and the only information contained in these patterns is where the variance of the EEG varies most when compared to the two conditions.

Considering the N number of EEG channels for each class 1 and class 2 and T samples for each channel, the dimension of the data matrix is N × T. The normalized covariance matrix for each class is:(1)$$C_{1}=\frac{X_{1}\times X_{1}^{T}}{trace\left(X_{1}\times X_{1}^{T}\right)}$$
(2)$$C_{2}=\frac{X_{2}\times X_{2}^{T}}{trace\left(X_{2}\times X_{2}^{T}\right)}$$

The transpose of the matrix *X* is *X^T^*, while the sum of the elements of the major diagonal of matrix A is trace(.). ${\bar{C}}_{1}$ and ${\bar{C}}_{2}$’s averaged normalized covariances are calculated by averaging the normalized covariances across all trials [35]. The combination of these covariance matrices can be decomposed as follows:(3)$$C={\bar{C}}_{1}+{\bar{C}}_{2}=U_{0}U_{0}^{T}$$
where $U_{0}$ denotes the eigenvectors and Σ is a diagonal matrix of eigenvalues of the covariance matrix C. After that, the average covariance matrices for both classes are converted ($P=\Sigma^{-1/2}U_{0}^{T}$) as follows:(4)$$S_{1}=P\bar{C_{1}}P^{T},\;S_{2}=P\bar{C_{2}}P^{T}$$

The eigenvectors for $S_{1}$ and $S_{2}$ are the same, and the combination of the related eigenvalues for both classes is one. To put this another way, it can be written as:(5)$$S_{1}=P\bar{C_{1}}P^{T},\;S_{2}=P\bar{C_{2}}P^{T}$$

As a result, the $S_{1}$ eigenvectors have the highest value, while the $S_{2}$ eigenvectors have the lowest value, and vice versa. As a result, the weights matrix is as follows:(6)$$W=U^{T}$$

The eigenvalues matrix is sorted in descending order in this algorithm. As a result, the first m and last m weights, which correspond to the greatest and minimum values, are picked, while the remaining weights are left as zeros [35]. EEG data can be transformed into distinct components in this way, such as:(7)Z=WX 

The variance of *X* is projected onto the rows of *Z* using this transformation, yielding N new time series. *W*^−1^’s columns are a collection of CSPs that can be thought of as time-invariant EEG source distributions.

According to *W*’s definition, the variance for class 1 assigned movement imagination is greatest in the first row of *Z* and decreases as the number of following rows increases. In an experiment involving a different type of motor imagery, the reverse occurs. The variances must be extracted as reliable features of the newly created N time series to categorize class 1 versus the other classes, for example. It is not required, however, to compute the variances of all N time series. The EEG’s dimensionality is reduced using this procedure. Mueller-Gerking et al., [36] found that four common spatial patterns is the ideal number. Following their findings, only the first and last two rows (*p* = 4) of *W* are used to process new input data *X* after creating the projection matrix *W* from an artifact corrected training set *X^T^* [37]. Thus, four features result from spatially filtering the EEG data [16]. These four features are further preprocessed before classification is done. First, the variance (*VARp*) of the resulting four-time series is calculated for a 1.5 s window.
(8)$$VAR_{p}=\sum_{t=1}^{T}{\left(Z_{p\left(t\right)}\right)}^{2}$$

Secondly, they are normalized and log-transformed, obtaining four feature vectors.
(9)$$f_{p}=log\frac{VAR_{p}}{\sum_{p=1}^{4}VAR_{p}}$$

To discriminate between 3 classes, we applied the principle of one versus all, meaning that, for each class, we calculated the CSP filters and classifiers for that class versus the others as described in the first part of this section. In this way, for each class, we calculated 4 sets of CSPs. With these four features, a Fisher’s linear discriminant analysis (LDA) [38], classification was performed to categorize the movement either as that class, or the other classes.

From the training data recorded during the first four runs, multiple sets of spatial filters and classifiers were off-line calculated from two, three, and four seconds time windows shifted in time with 0.5 s in the 4–8 s trials time interval. The classifier with the highest ten-fold cross-validated accuracy [39] was chosen to be used to provide visual and FES feedback while recording runs five and six. While performing the first four runs recordings, the feedback was provided using the spatial filters and classifiers achieved in the previous session.

### 2.4. Off-Line Data Processing

During the offline data processing, all recorded trials were visually inspected for artifacts in the time windows between 3 to 8 s. Trials containing artifacts were excluded because of the sensitivity of the CSP method to artifacts. The reason is the sample covariance (non-robust estimate), which is used to estimate the covariance for the calculation of the spatial filters. While BCI operates online, the spatial filters perform a weighted spatial averaging of the EEG, and this reduces the influence of artifacts [16]. The spatial patterns and classifiers from the merged runs (1,2,3,4) were computed using three different window lengths (2 s, 3 s, and 4 s). These windows were moved forward in time by 0.5 s to cover the whole feedback phase (see Figure 5). By comparing the provided cue with the classified movement on runs 5 and 6, an error rate for each set has been calculated. Signal fractions of half a second were used to compute each classifier and error. The classifiers were applied to the features for each fraction, and the classification results were compared to the cue, yielding error rates that were averaged over all trials [40].

### 2.5. Robot Control Unit

For the robotic application, a 6 degree of freedom industrial robot arm ABB IRB120 was used, which is shown in Figure 6. The robot computer communicates with the BCI system over the TCI/IP communication protocol. ABB IRB 120 is a lightweight robot and has a 3-kg payload capacity. It can reach 580 mm away from its center point. Position repeatability of the robot arm is 0.01 mm. The speed of the manipulator was set at 100 mm/s in each direction. An IRC5 controller was used for controlling the robot arm.

To properly control the robot manipulator, the RAPID programming language was used. The main goal of this program was to receive commands which had been sent from the BCI system and to generate corresponding movement functions in real-time. In this study, the motor imagery-based BCI system generates three specific control signals (left, right, and forward). These control signals were received by the robot controller and converted to three directional move commands as shown in Figure 7.

## 3. Results

After finishing the recording for all subjects, we continued off-line testing the influence of different variance time windows on the classification result. We repeated the CSP and classifier generation presented in section off-line data processing for each of the 1, 1.5, and 2 s variance time windows. Then, we generated the results offline by applying the created spatial filters and classifiers on the last two runs of data from each subject. To perform a better comparison for all subjects we calculated the mean error from the feedback time interval between seconds 5 and 8. The mean and minimal errors for all subjects are presented in Appendix A.

For all subjects, the lowest mean errors were achieved for a 3-s CSP time window and 1.5 s variance smoothing time. To illustrate, Figure 8 presents the results of applying CSP filters and classifier created from seconds 3.5–6.5 (delimitated in the left plots by two vertical dashed lines), for a 1.5 s variance time on the last two runs for subject 1. The dashed lines in the left-side plots represent the LDA classifier output for each right hand, left hand, and feet movement imagination trial. The solid lines are averages of each set of trials for every time point. The plot on the right side presents the online classification error rate for the last 2 runs. Figure 9, Figure 10 and Figure 11 present the CSP maps (right versus all, left versus all, and foot versus all) of subject 1. The white points in the figures show the positions of the 63 electrodes. A cross was used to denote points of C3 and Cz and C4. Among the 63 CSPs, the first and the last were the most and the second and sixty-second were the second most discriminating filters in each case. The first pattern in Figure 9 shows a blue spot in the region of C4 (controlling the left hand), the first pattern in Figure 10 shows a red spot in the region of C3 (controlling the right hand), and the first pattern in Figure 11 shows a red spot at Cz site (foot movement imagery) [40].

## 4. Discussion

This study has attempted to optimize the configuration parameters of an MI-based BCI designed for robotic arm control. To do so, the classification accuracy of a set of 12 different time windows of data for creating the spatial filters and classifiers was analyzed. At the same time, we also evaluated three different variance time windows of 1 s, 1.5 s, and 2 s. Based on the data, we averaged across the error values achieved for all CSP and variance time windows (Table 1).

The lowest error rates for both mean and minimal errors are bold-written in Table 1. Regarding the variance smoothing time, the 1.5 s time window yielded in four of seven mean error cases the lowest classification error rates. When it came to the CSP time window construction, in general, the longer time windows generated higher accuracy levels. When averaging overall findings (see Table 2), the best error rate was for 3 s-length time window. The larger the time window for CSP production, the more critical it is to do a precise artifact correction. This means that if you take a window of only 2 s, only artifacts within that second are relevant; if you take a window that covers almost the entire feedback period, the entire time must be free of artifacts. The fact that the 4-s time window performed significantly worse in some cases than the 3-s time window could be due to the fact that they begin at second 3.5 and 4, right after the cue presentation. However, the EEG only begins to provide discriminative signals for the MI task at this point.

As a result, starting the window period later is suitable. Based on these results, we decided to use a 3 s window for CSP filter creations and a 1.5 s variance smoothing time for further studies.

During the recording sessions, with almost every set of CSP and weight vector (WV), S1 and S3 reached mean error rates below 15%. For both S1 and S3, with a single exception, the minimal error was below 8%, which means a BCI control accuracy of more than 92%. The error level for the other subjects was poorer, in comparison to S1 and S3, but still acceptable, yielding a control accuracy of about 70%. More training might have enhanced the accuracy of these subjects positively. The results attained in this case are similar to the results reported by other authors, Ridha Djemal et al., (2016), who used the Fast Fourier Transform (FFT) and autoregressive (AR) modeling of the reconstructed phase space for classifying the data [41].

Even if the classification results and commands sent to the robot were pretty accurate, particularly for S1 and S3, the number of 3 commands that are sent to the robot is limited in the present study. As follow-up research, we plan to design a hybrid BCI controlling system, which will also embed the SSVEP mental strategy. The SSVEP will be used also as a switch between multiple classes of commands for the robot, and the motor imagery for fine-tuning the movements as the SSVEP strategy offers the advantage of shorter calibration time.

## Figures and Tables

**Figure 1 brainsci-12-00833-f001:**
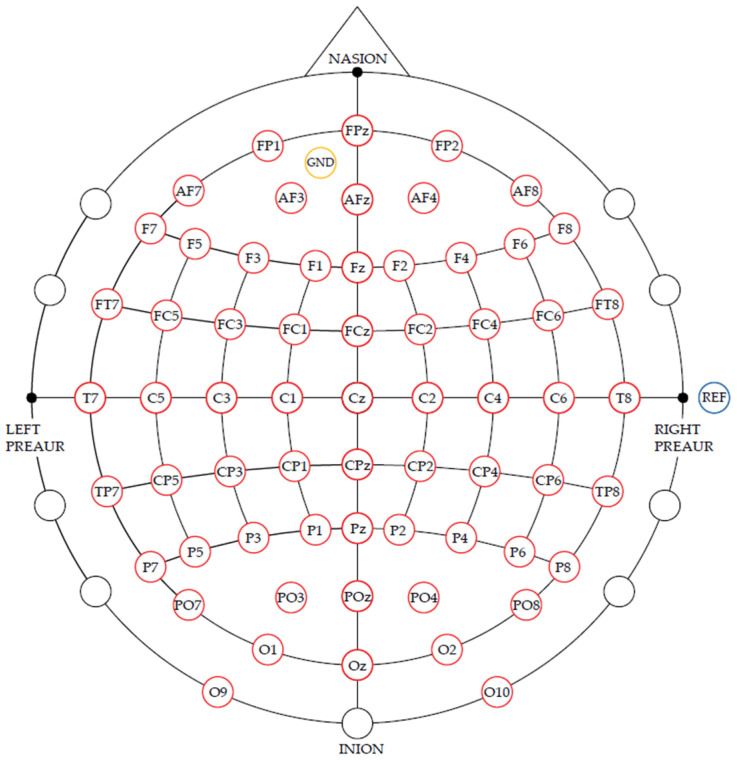
64 EEG electrodes placed on the scalp according to the 10-10 international system.

**Figure 2 brainsci-12-00833-f002:**
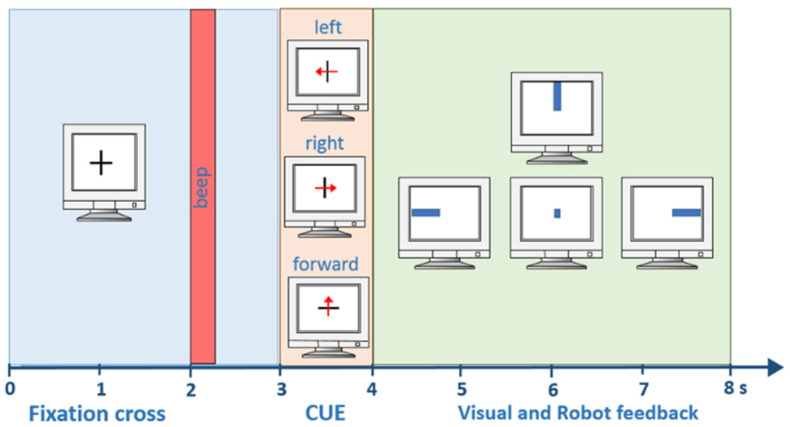
Timing within one trial.

**Figure 3 brainsci-12-00833-f003:**
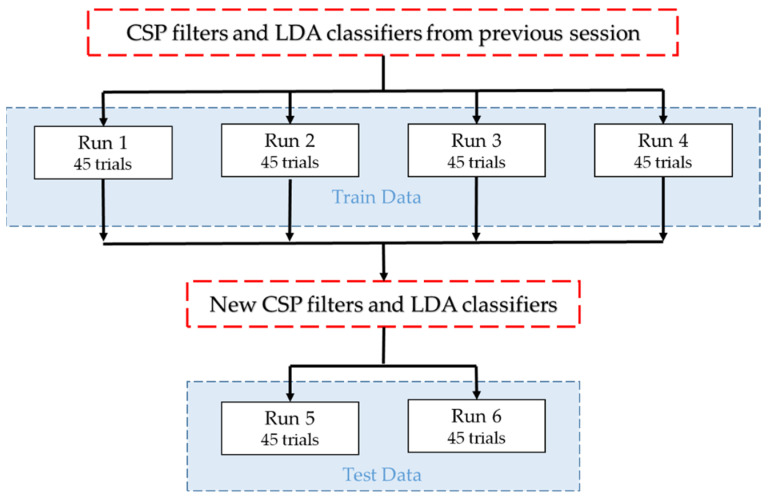
Flow diagram of one session: training data (run1–4) are recorded by using CSP filters and the classifier from the previous session. Training data was used to generate new CSP filters and a classifier. Test data (run5, 6) were recorded by using new filters and a classifier [34].

**Figure 4 brainsci-12-00833-f004:**
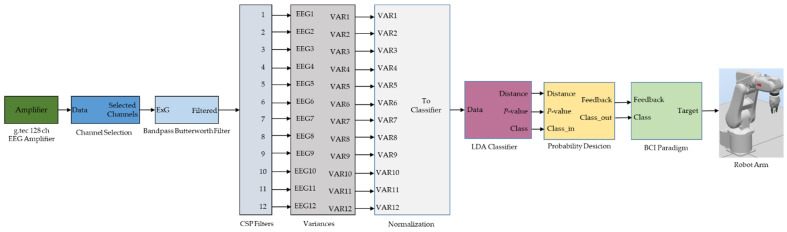
Motor imagery-based BCI model to control a robot arm.

**Figure 5 brainsci-12-00833-f005:**
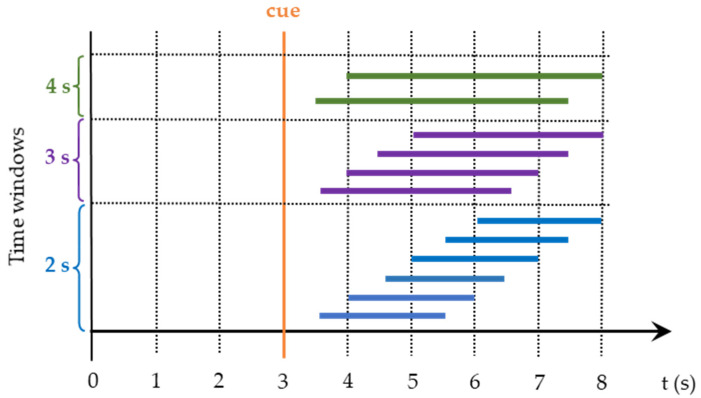
To calculate CSP filters and classifiers different time window–length segments were selected.

**Figure 6 brainsci-12-00833-f006:**
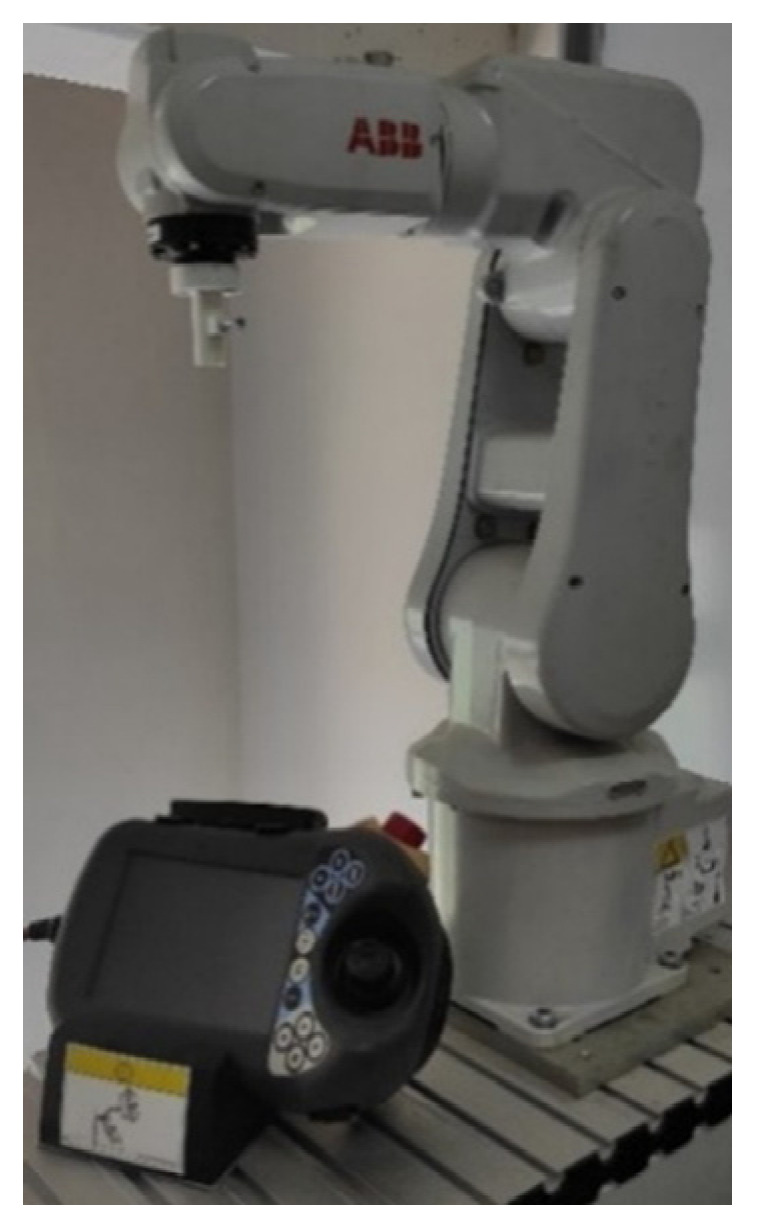
6-degree of freedom industrial robot arm.

**Figure 7 brainsci-12-00833-f007:**
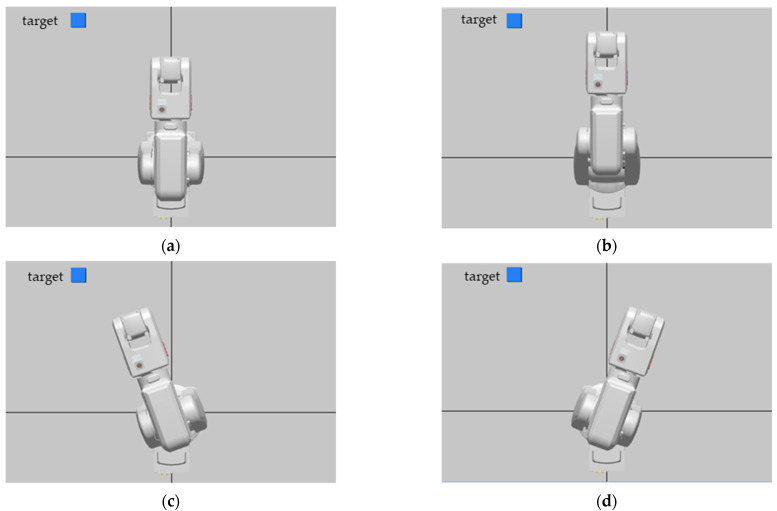
(**a**) Home position of the robot arm, (**b**) forward movement, (**c**) left movement, (**d**) right movement.

**Figure 8 brainsci-12-00833-f008:**
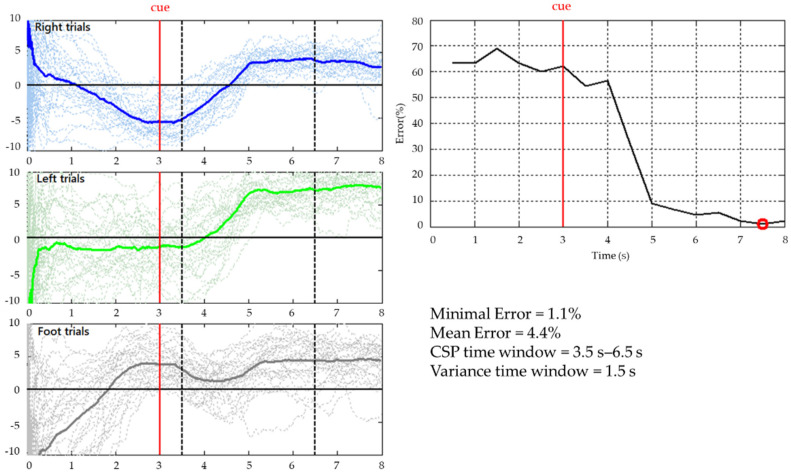
Classification results for subject 1. Left–side plots: LDA classification output for right, left, and foot imagination tasks; right–side plot: Online classification error rate.

**Figure 9 brainsci-12-00833-f009:**
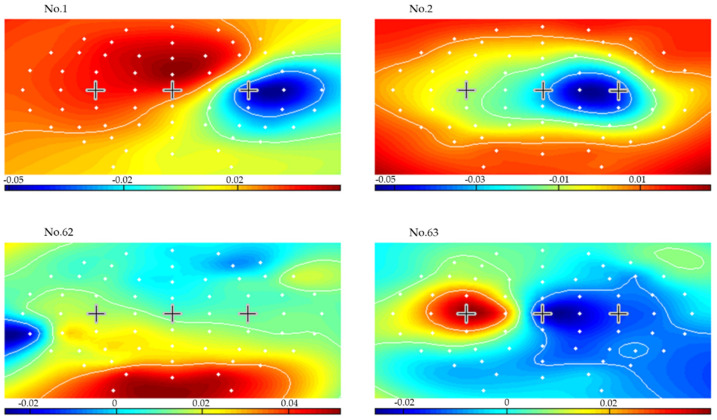
Maps of the most significant common spatial patterns for a 3 s window length from second 3.5 to second 6.5 for left hand versus all (subject 1).

**Figure 10 brainsci-12-00833-f010:**
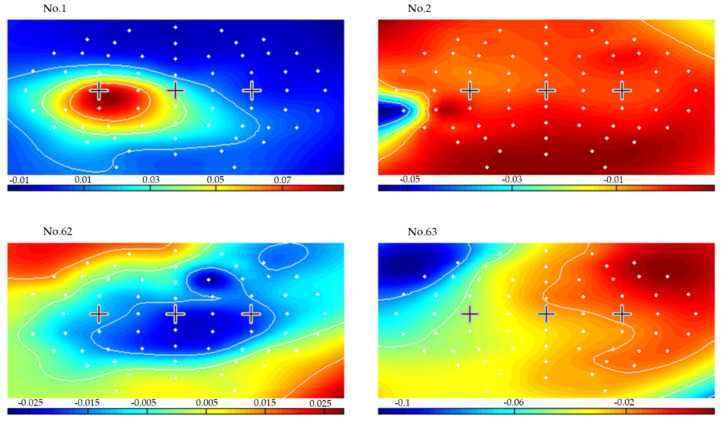
Maps of the most significant common spatial patterns for a 3 s window length from second 3.5 to second 6.5 for right hand versus all (subject 1).

**Figure 11 brainsci-12-00833-f011:**
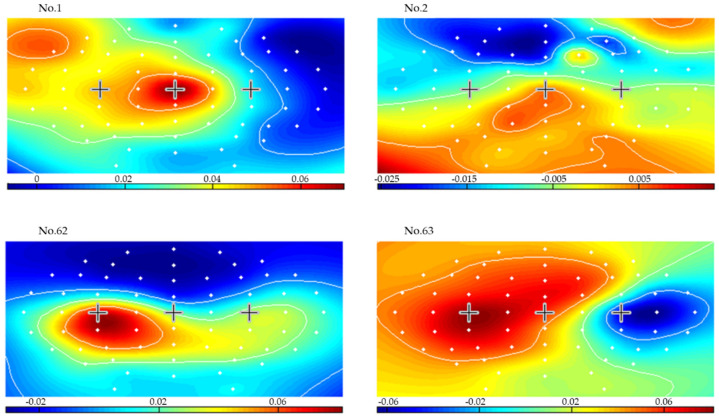
Maps of the most significant common spatial patterns for a 3 s window length from second 3.5 to second 6.5 for feet versus all (subject 1).

**Table 1 brainsci-12-00833-t001:** Error-values vs. variance time windows for all CSP.

		Mean Errors Average			Min Errors Average		
Subject	Variance Time Window	2 s	3 s	4 s	2 s	3 s	4 s
**S1**	1 s	10.15	10.08	9.30	6.60	6.70	6.70
	1.5 s	7.12	**4.98**	6.20	2.76	2.48	2.20
	2 s	7.28	7.33	5.80	1.65	2.48	**0.55**
**S2**	1 s	35.04	33.83	34.20	29.27	28.08	27.80
	1.5 s	33.99	**32.58**	34.45	27.97	**26.40**	29.45
	2 s	33.68	33.23	33.25	26.94	26.65	27.25
**S3**	1 s	6.93	6.73	6.35	3.77	3.78	3.80
	1.5 s	6.62	6.10	**6.05**	1.68	1.58	2.50
	2 s	7.80	7.63	8.70	2.10	1.90	**1.30**
**S4**	1 s	**37.30**	40.78	46.40	**33.55**	37.53	43.75
	1.5 s	37.58	40.03	47.90	34.13	36.60	46.30
	2 s	38.17	41.35	48.00	34.62	36.60	46.30
**S5**	1 s	16.02	12.10	8.75	10.05	5.65	2.50
	1.5 s	12.32	8.40	8.75	5.87	2.83	2.50
	2 s	10.85	6.65	**5.85**	5.45	2.23	**1.90**
**S6**	1 s	40.95	41.95	41.00	**31.67**	33.45	36.25
	1.5 s	41.48	40.68	39.45	33.57	33.15	36.25
	2 s	40.72	40.93	**39.30**	33.77	35.35	35.65
**S7**	1 s	13.70	11.40	11.25	7.95	7.20	7.50
	1.5 s	10.93	10.38	**9.90**	6.25	7.55	6.90
	2 s	10.78	10.05	10.45	5.85	**5.65**	8.15

**Table 2 brainsci-12-00833-t002:** Grand averages for mean and minimal errors presented in Table 1.

	Grand Average of Mean Errors			Grand Average of Minimal Errors		
Variance Time Window	2 s	3 s	4 s	2 s	3 s	4 s
1 s	22.87	22.41	22.46	17.55	17.48	18.33
1.5 s	21.43	**20.45**	21.81	16.03	15.80	18.01
2 s	21.33	21.02	21.62	**15.77**	15.84	17.30

## Data Availability

Not applicable.

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
