# Peer review of "Optimizing Motor Imagery Parameters for Robotic Arm Control by Brain-Computer Interface"

_brainsci, 2022, doi:10.3390/brainsci12070833_

Round 1
Reviewer 1 Report
This paper studies an important problem that uses BMI to control a robot arm. The system is clearly described. Some concerns still remain:
- The introduction part gave a very detailed introduction of different types of BMI. However, this is not the main contribution of this paper. More detailed and specific literature review is required.
- For the feature extraction, it is not clear the exact algorithm that used.
- For the control of the robot arm, there are already many relevant works that use BMI to control the robot arm. What is new here?
Author Response
Dear Sir/Madam,
The Revised manuscript of the paper (brainsci-1721667) “Optimizing Motor Imagery Parameters for Controlling of Robotic Arm by Brain-Computer Interface” has been prepared accordingly regarding the recommendations of the reviewers. The length of the revised paper has been decreased by moving 4 figures to the supplementary material, as recommended by the reviewers. In addition, the manuscript was proofread and corrected by an English native speaker. Based on his/her suggestions we also modified the article title to “Optimizing Motor Imagery Parameters for Robotic Arm Control by Brain-Computer Interface”.
The authors’ responses to the reviewer’s comments are presented in detail as follows:
This paper studies an important problem that uses BMI to control a robot arm. The system is clearly described. Some concerns still remain:
The authors thank the reviewer for the comments. First of all, we asked an English native speaker to proofread the manuscript and to help us with the grammar modifications. Based on his/her advice, we changed the manuscript title to “Optimizing Motor Imagery Parameters for Robotic Arm Control by Brain-Computer Interface”.
- The introduction part gave a very detailed introduction of different types of BMI. However, this is not the main contribution of this paper. More detailed and specific literature review is required.
We added to the Introduction section a new paragraph about robotic arms and assistive devices, together with some new references.
- For the feature extraction, it is not clear the exact algorithm that used.
The feature extraction algorithm uses the standard CSP method that we described at the beginning of section 2.3. We added a sentence in the final part of the section clarifying this issue.
- For the control of the robot arm, there are already many relevant works that use BMI to control the robot arm. What is new here?
We described in the second part of the Introduction section that it is very important to find a way (algorithm or parameters) to get the lowest classification error rate, because usually disabled people gain lower BCI performance due to their condition, highlighting our method of choosing the optimal parameters for this kind of mental strategy.

Reviewer 2 Report
The paper presents an interesting study on the optimization of the configuration parameters of a BCI-based robotic arm. I value the motivation of the work, but the overall presentation was low quality. I have two main comments:
- I would suggest that the authors seek help from professional English writers. You do not need to sound like native speakers, but there are too many grammar errors in the manuscript. Language proofreading is needed here. Or at least check your grammar issue using free software e.g., grammarly.
- Re-organize the figures. A paper of 14 figures can be overwhelming to your readers. Combine some of the figures and put the other into supplementary materials. Try to keep the most important data in the main text to make your manuscript concise and focused.
Author Response
Dear Sir/Madam,
The revised manuscript of the paper (brainsci-1721667) “Optimizing Motor Imagery Parameters for Controlling of Robotic Arm by Brain-Computer Interface” has been prepared accordingly regarding the recommendations of the reviewers. The length of the revised paper has been decreased by moving 4 figures to the supplementary material, as recommended by the reviewers. Moreover, the manuscript was proofread and corrected by an English native speaker, and based on his suggestions we modified also the article title to “Optimizing Motor Imagery Parameters for Robotic Arm Control by Brain-Computer Interface”.
The authors’ responses to the reviewer’s comments are presented in detail as follows:
The paper presents an interesting study on the optimization of the configuration parameters of a BCI-based robotic arm. I value the motivation of the work, but the overall presentation was low quality. I have two main comments:
- I would suggest that the authors seek help from professional English writers. You do not need to sound like native speakers, but there are too many grammar errors in the manuscript. Language proofreading is needed here. Or at least check your grammar issue using free software e.g., grammarly.
The authors thank the reviewer for the comments. We asked an English native speaker to proofread the manuscript and to help us with the grammar modifications. Based on his/her advice, we changed the manuscript title to “Optimizing Motor Imagery Parameters for Robotic Arm Control by Brain-Computer Interface”.
- Re-organize the figures. A paper of 14 figures can be overwhelming to your readers. Combine some of the figures and put the other into supplementary materials. Try to keep the most important data in the main text to make your manuscript concise and focused.
We moved 4 figures from the manuscript to the Supplementary material file.

Reviewer 3 Report
This study intends to optimize the classification of motor imagery trials using a robotic arm. In my opinion, the investigation of different time windows for trial classification and spatial filtering is rather new and interesting. In general, methods are well described.
Please find below a couple of comments:
- Labels in figure 1 are not visible
- Figure 4 is not visible.
- Consider renaming "time window" into "epoch".
- when it comes to covariance matrices, it is in general expected to mention Riemannian geometry.
- The authors used the CSP algorithm. May be worth mentioning also CSTP?
Author Response
Dear Sir/Madam,
The revised manuscript of the paper (brainsci-1721667) “Optimizing Motor Imagery Parameters for Controlling of Robotic Arm by Brain-Computer Interface” has been prepared accordingly regarding the recommendations of the reviewers. The length of the revised paper has been decreased by moving 4 figures to the supplementary material, as recommended by the reviewers. Moreover, the manuscript was proofread and corrected by an English native speaker, and based on his suggestions we modified also the article title to “Optimizing Motor Imagery Parameters for Robotic Arm Control by Brain-Computer Interface”.
The authors’ responses to the reviewer’s comments are presented in detail as follows:
This study intends to optimize the classification of motor imagery trials using a robotic arm. In my opinion, the investigation of different time windows for trial classification and spatial filtering is rather new and interesting. In general, methods are well described.
Please find below a couple of comments:
The authors thank the reviewer for the comments. First of all, we asked an English native speaker to proofread the manuscript and to help us with the grammar modifications. Based on his/her advice, we changed the manuscript title to “Optimizing Motor Imagery Parameters for Robotic Arm Control by Brain-Computer Interface”.
- Labels in figure 1 are not visible.
- Figure 4 is not visible.
We modified figures 1 and 4 to be more visible.
- Consider renaming "time window" into "epoch".
We modified time-window to epoch in a couple of places, but not in all places. In general, the epoch refers to a whole trial. Our time window consists of a number of samples. The window that is shifted in time belongs to an epoch.
- when it comes to covariance matrices, it is in general expected to mention Riemannian geometry.
We added to the introduction the CSP method based on Riemannian geometry together with a reference. In our method, we use the standard CSP method that uses only the first 2 and last 2 CSPs. Indeed, the CSP based on Riemannian geometry is an interesting technique that allows to choose the optimal number of CSPs and the specific ones. We thank you for this suggestion, and we will try it in our future studies.
- The authors used the CSP algorithm. May be worth mentioning also CSTP?
We also mentioned the CSTP extension in the introduction with a corresponding reference.
